# Generalizable Geometric Image Caption Synthesis

## Abstract

Multimodal large language models (MLLMs) have various practical applications that demand strong reasoning abilities. Despite recent advancements, these models still struggle to solve complex geometric problems. A key challenge stems from the lack of high-quality image-text pair datasets for understanding geometric images. Furthermore, most symbolic data synthesis pipelines typically fail to generalize to questions beyond their predefined templates. In this paper, we bridge this gap by introducing a complementary process of Reinforcement Learning with Verifiable Rewards (RLVR) into the data generation pipeline. By adopting accuracy-guided RLVR to refine captions for symbolically synthesized geometric images, our pipeline successfully captures the key features of geometry problem-solving. This enables better task generalization and yields non-trivial improvements. Furthermore, even in out-of-distribution scenarios, the generated dataset GeoReasoning-10K achieves non-trivial performance gains, yielding accuracy improvements of 2.8%–4.8% in non-geometric subtasks of MathVista and MathVerse. This generalization ability is further validated in MMMU, where significant improvements of 2.4%–3.9% in Art & Design and Tech & Engineering tasks are observed.

## 1 Introduction

Multimodal Large Language Models have exhibited impressive capabilities across a variety of vision-related tasks, including Visual Question Answering (VQA), visual grounding, and image captioning. Recent MLLMs, such as Qwen2.5-VL, Intern2.5-VL, and LLaVA-Next (Bai et al., 2025; Chen et al., 2024; Liu et al., 2024), have shown superior performance compared to specialized vision models across a wide range of visual tasks. As the field advances, there has been increasing interest in enhancing the reasoning capabilities of MLLMs (Jaech et al., 2024; Shao et al., 2024), which is regarded as a crucial factor in extending the performance boundaries of these models. Among various reasoning tasks, mathematical reasoning (Zhang et al., 2024a) has attributed particular attention due to its structured problem-solving nature, offering a clear pathway for MLLMs to develop and improve their reasoning skills.

Research from MathVerse (Zhang et al., 2024a) indicates that MLLMs perform best when the input is purely textual, while their performance declines significantly with visual-only inputs. This highlights the urgent need for MLLMs to develop strong cross-modal reasoning capabilities, which involves accurately and comprehensively transferring information from the image to the text. Although numerous geometry and math datasets have been introduced (Lu et al., 2023; Zhang et al., 2024b; Wang et al., 2024) to boost various aspects of model performance, high-quality datasets explicitly designed for cross-modal reasoning remain scarce. That is primarily because in existing datasets, the alignment between images and captions is often asymmetrical. For instance, in geometric problems, two lines of equal length can be easily described textually but may not be correspondingly annotated or visually distinct in the image. Such discrepancies hinder the model's ability to learn robust cross-modal reasoning.

Meanwhile, Reinforcement Learning (RL) has been demonstrated to significantly improve model reasoning and generalization capabilities (Guo et al., 2025). Its reward-driven framework is particularly effective for cross-modal reasoning, allowing models to optimize decision-making through interactive feedback (Deng et al., 2025; Peng et al., 2025; Huang et al., 2025a). Building on these insights, we

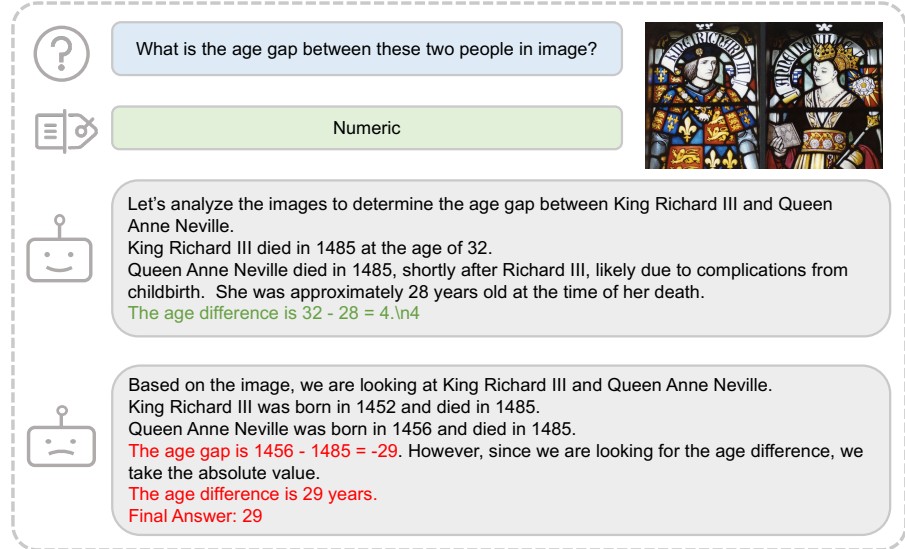

Figure 1: **Examples of generalization**. MLLMs learn from our synthetic geometric mathematical problems and generalize to algebraic cases with even non-geometric input images.

employ the RAFT method (Dong et al., 2023) and design a reward function that incorporates both reasoning and caption rewards. This facilitates the alternating optimization of dataset quality and model reasoning abilities, leading to improved results on complex multimodal tasks.

To bridge the gap between visual and linguistic modalities, we propose an RL-based data refinement engine that iteratively enhances data quality. Utilizing this pipeline, we introduce a novel geometry dataset comprising 10,000 image-caption pairs. To the best of our knowledge, this is the first high-quality dataset in which visual and textual information are fully aligned and generalize well to out-of-distribution tasks, making it a valuable resource for improving cross-modal reasoning. Experimental results demonstrate that our dataset significantly enhances models' cross-modal reasoning abilities and their performance on geometric image textualization tasks. Furthermore, models trained on our dataset exhibit strong generalization capabilities on other mathematics-focused benchmarks, including MathVerse and MathVista, as shown in Figure 1. In summary, our main contributions are summarized as follows:

- We introduce **GeoReasoning-10K**, a new dataset containing 10,000 carefully constructed image-caption pairs where visual and textual information are fully equivalent. This dataset serves as a high-quality resource for training cross-modal reasoning models, outperforming previous geometric datasets like AutoGeo in both MathVerse and MathVista.

- We propose **Geo-Image-Textualization**, a scalable RL-based framework for generating and refining high-quality synthetic image-caption pairs in geometry. Our iterative RL-driven optimization significantly enhances data alignment and semantic accuracy, and demonstrates generalization to out-of-domain geometric tasks.

- Extensive experiments show that the improvements facilitated by GeoReasoning extend beyond geometric tasks and even generalize well to non-mathematical domains. In particular, GeoReasoning-10K brings an accuracy improvement of **+2.8%–4.8%** in non-geometric subtasks of MathVista and MathVerse, and **+2.4%–3.9%** in Art & Design and Tech & Engineering tasks in MMMU.

## 2 RELATED WORKS

### 2.1 DATA GENERATION

Recent studies have highlighted the scarcity of high-quality geometry image–caption datasets, which limits fine-grained cross-modal reasoning in geometric tasks. Following AlphaGeometry (Trinh

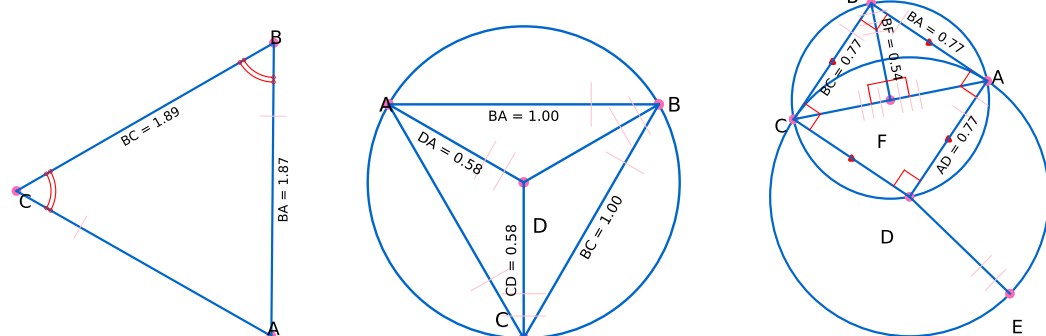

Figure 2: **Symbolically synthesized geometric images**. These geometry problems are symbolically composed from our relation library, corresponding to easy, medium, and hard difficulty levels, respectively, where the pink ticks and red arcs indicate equal-length segments and equal angles. The symbolic engine can generate images with infinite types and difficulties. For visual clarity, this figure has a fixed set of colors, font sizes, and line thicknesses compared to the original images in our constructed dataset. Please refer to the original dataset for precise details.

et al., 2024), Autogeo (Huang et al., 2025b) proposed an automatic generation engine to produce image-caption pairs, constructing a 100K dataset named AutoGeo-100k. MATHGLANCE (Sun et al., 2025) introduced GeoPeP, a perception-oriented dataset of 200K structured geometry image-text pairs explicitly annotated with geometric primitives and spatial relationships. MagicGeo (Wang et al., 2025) formulates diagram synthesis as a coordinate optimization problem, ensuring formal geometric correctness via solvers before coordinate-aware text generation.

Despite the advances, existing pipelines still struggle to guarantee full modality alignment, i.e., captions frequently omit visual details, while images lack exhaustively aligned textual descriptions.

## 2.2 IMAGE CAPTIONING

Image captioning aims to generate comprehensive descriptions of visual content and has been widely studied for natural images. While general-purpose MLLMs such as mPLUG-Owl2 (Ye et al., 2024), MiniGPT-4 (Zhu et al., 2023), and BLIP-3 (Xue et al., 2024) can perform captioning to some extent, their effectiveness is often limited by insufficient fine-grained cross-modal alignment. Image-Textualization (Pi et al., 2024) mitigates this issue by integrating multiple vision experts to produce more detailed and accurate captions.

However, the potential of utilizing image captioning to enhance geometric reasoning capacity remains underexplored. OmniCaptioner (Lu et al., 2025) proposes a unified visual captioning framework that converts diverse images into fine-grained textual descriptions. Nonetheless, its geometric annotations are derived from AutoGeo and MAVIS, largely relying on synthetic or loosely aligned pairs rather than fully equivalent visual-textual representations. Moreover, the scarcity of high-quality geometric image-caption pairs makes it difficult to accurately extract and align geometric information. As a result, current models underperform on geometric image textualization compared to natural image captioning and general visual reasoning benchmarks.

## 3 METHODS

In this section, we introduce our Geo-Image-Textualization data generation pipeline first, followed by our RAFT method used for data refinement.

### 3.1 GEO-IMAGE-TEXTUALIZATION DATA GENERATION ENGINE

The proposed data generation pipeline mainly contains three parts: the relation sampling, image-caption pair generation, and question-answer generation procedure, as shown in Figure 3.

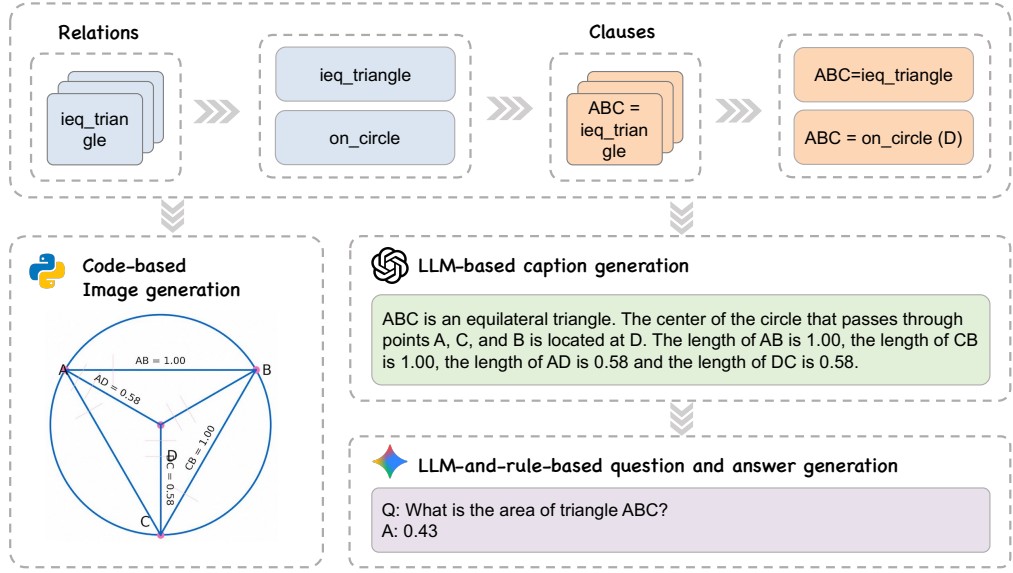

Figure 3: **The geometry data synthesis pipeline**, where a graph-based representation similar to AutoGeo (Huang et al., 2025b) is employed for generating the final geometry images. The relation library comprises over 50 basic geometric relationships that can be composed into complex ones, providing comprehensive coverage for geometric problems of various difficulties. The image-caption pair is utilized for the SFT stage, while the caption-QA pair for the RLVR stage.

### 3.1.1 RELATION SAMPLING

We develop the Geo-Image-Textualization pipeline upon the data generation procedure in Alpha-Geometry. In our framework, *Relations* act as fundamental construction operations that systematically generate diverse yet semantically coherent geometric premises for subsequent theorem synthesis. Each relation (e.g., angle_mirror, circumcenter, etc.) encodes a precise geometric procedure—such as reflecting a point across an angle bisector or locating the circumcenter of a triangle. In addition to the construction rule, each definition maintains dependency metadata, specifying which primitive objects (points, lines, circles) and prior constructions it depends on. This enables the symbolic engine to get the minimal set of premises required to derive a given theorem.

After sampling relations, each relation is converted into a clause, with associated point variables. For instance, the relation angle_mirror x a b c denotes: given points a, b, and c, construct point x as the reflection of c across angle $\angle ABC$. Finally, the system constructs a graph-based representation in AutoGeo (Huang et al., 2025b) to model geometric problems. Each clause, corresponding to either a geometric construction or a relational assertion, is incorporated into the graph by instantiating nodes for geometric entities (e.g., points, lines, circles) and establishing edges that encode their interdependencies. Before adding each clause, the system verifies the logical correctness of the selected set of predefined geometric rules.

### 3.1.2 IMAGE-CAPTION PAIR GENERATION

The geometric graph encodes all relevant entities, including points, lines, and circles, enabling the straightforward rendering of basic geometric elements, similar to AutoGeo. However, a fundamental limitation of AutoGeo is that the captions cannot be directly inferred from the rendered images because the visual content and the textual description are not semantically aligned. We argue that this misalignment constitutes a critical bottleneck in cross-modal reasoning.

To address this issue, we introduce a set of visual augmentation strategies that explicitly encode semantic relationships within the image, following most conventions in geometry problems (Dimmel & Herbst, 2015):

1. **Segment Equality Representation:** We use short perpendicular ticks to indicate equal-length line segments. When multiple pairs of equal segments exist, we distinguish them using a different number of ticks (e.g., one tick, two ticks)[1].

2. **Angle Annotations:** For angles that are integer multiples of $15°$ within the range $[15°, 165°]$, we directly annotate the angle values within the image.

3. **Parallel and Perpendicular Indicators:** Parallel lines are marked using matching directional triangles, and right angles are indicated using a small square symbol at the vertex.

4. **Equal Angle Representation:** Equal angles are denoted by marking them with the same number of arcs, consistent with conventional geometric notation.

5. **Intersection and Collinearity:** Dashed lines are used to explicitly indicate intersections and collinearity relationships among points or segments.

For each clause in the symbolic representation, we apply a refined, rule-based template to convert it into natural language. These captions accurately describe the geometric diagram, including object relationships, special angle values, and other visual properties. Additionally, the captions explicitly state the lengths of specific line segments when such information is visually annotated in the image. By ensuring that all semantic content present in the image is mirrored in the caption, we achieve full cross-modal alignment.

## 3.2 QUESTION-ANSWER PAIR GENERATION

The most fundamental requirements for generating questions lie in three aspects. First, the generated question should be based on the caption, i.e., should not be irrelevant to the caption. Second, any information already present in the caption should be removed, as this would dilute the impact of the caption and make the evaluation of caption quality harder. Last, the question should be compatible with the given information, so that it can be logically answered based on what is provided.

Based on these requirements, we propose a rule-and-LLM-based pipeline to systematically generate the question and answer based on the pre-generated caption. Specifically, we prompt the large model (Gemini 2.5 Flash) with rubric-based instructions to generate initial questions based on the caption, while also letting the model flag those questions inconsistent with the caption. For the inconsistent questions, we then switch to a different prompt, encouraging the model to incorporate additional information and formulate new questions accordingly. This process continues until a self-consistent question is generated for the first time. The detailed two-stage prompt design is provided in Appendix A and B.

## 3.3 RLVR FRAMEWORK FOR DATA REFINEMENT

Our proposed RLVR framework iteratively optimizes both the model and the dataset through a novel alternating paradigm. The method consists of two phases: (1) a cold-start supervised fine-tuning phase to bootstrap initial captioning capabilities, and (2) an RLVR phase with RAFT (Dong et al., 2023) that cyclically refines the dataset and model via reinforcement learning. The overall framework is shown in Figure 4.

### 3.3.1 COLD-START PHASE

To initialize the model's ability to generate geometrically aligned captions, we first perform supervised fine-tuning (SFT) on the base vision language model using the GeoReasoning-10K dataset. This phase minimizes the standard cross-entropy loss:

$$\mathcal{L}_{\text{SFT}} = -\mathbb{E}_{(I,c^\star)\sim\mathcal{D}_I}[\log P_{\theta_0}(c|I)] \tag{1}$$

where $I$ denotes an input geometric image, $c^\star$ and $c$ indicate the ground-truth caption and the predicted caption respectively, and $D_0$ represents the initial dataset. The model parameter $\theta_0$ is optimized to establish basic image-to-text mapping capabilities.

---

[1]This convention is similar to `https://mathbitsnotebook.com/Geometry/BasicTerms/BTnotation2.html`.

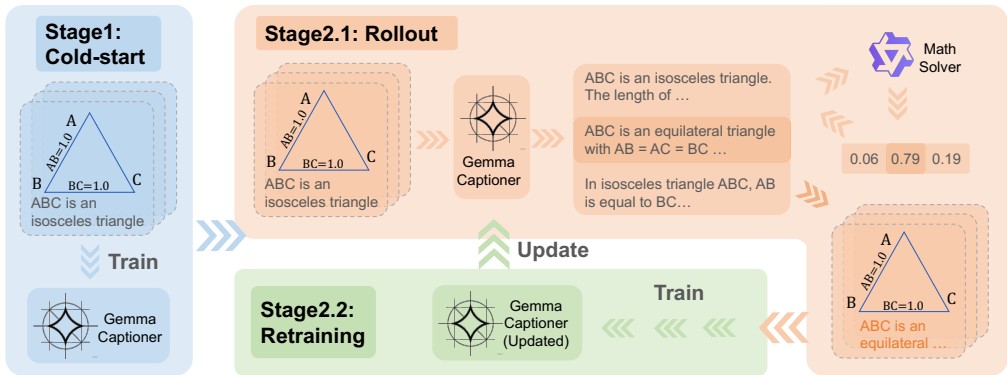

Figure 4: **The RLVR training framework**. In Stage 1, the model is trained to develop a preliminary ability to generate image captions. In Stage 2, an alternating optimization strategy is employed to jointly refine the generated captions and enhance the model's overall performance. The data of Stage 1 comes from the rule-based image-caption generation pipeline illustrated in Figure 3.

### 3.3.2 RLVR PHASE

The RLVR phase with RAFT operates in alternating stages, as shown in Figure 4.

**Rollout Experience Generation** Suppose the current iteration is $t$. For each image $I$ in the current dataset $\mathcal{D}_t$, we first generate $N$ candidate captions $\{c_i\}(i = 1, 2, \cdots, N)$ using the current vision language model with parameter $\theta_t$. Then, we utilize a specifically designed reward function $R(c_i, Q_i, c^\star)$ (detailed introduced in Section 3.3.3) to score each caption. Last, we retain the top-K caption $c_{\text{best}} = \arg\max_{c_i} R(c_i, Q_i, c^\star)$ to update the current dataset and construct the refined dataset $\mathcal{D}_{t+1}$.

**Model retraining** We update the model by training on $\mathcal{D}_{t+1}$ for one epoch, which is:

$$\theta_{t+1} = \arg\max_{\theta_t} \mathbb{E}_{(I, c_{\text{best}}) \sim \mathcal{D}_{t+1}}[\log P_{\theta_t}(c_{\text{best}}|I)] \tag{2}$$

This iterative process continues for $T = 5$ epochs, progressively enhancing both dataset quality and model performance.

### 3.3.3 REWARD FUNCTION

The composite reward $R(c, q, c^\star)$ balances task correctness and caption-image alignment, as shown in Eq. 3. And the overall reward modeling is shown in Figure 5.

$$R(c, I) = \lambda_r \cdot R_{\text{reasoning}}(c, q) + (1 - \lambda_r) \cdot R_{\text{caption}}(c, c^\star) \tag{3}$$

**Reasoning reaward** To evaluate a candidate caption's utility for solving downstream tasks, we leverage a frozen large language model of Qwen2.5-7B-Instruct (Yang et al., 2024) to generate an answer $a \sim P_{\text{LLM}}(a|q, a^\star, c)$ where $q, a^\star$ is the geometric question and its groundtruth answer generated by a reasoning model (Gemini2 .5 Flash) corresponding to the caption $c^\star$ in advance. As encouraged by mainstream RL process, we check both the format and correctness of the answer, which is:

$$R_{\text{Reasoning}} = s_c \cdot \mathbb{I}(a = a^\star) + (1 - s_c) \cdot \mathbb{F}(a) \tag{4}$$

where $\mathbb{F}(\cdot)$ denotes the format checking function, and $s_c$ indicate the weight of correctness, set as 0.9 in the experiments.

**Caption reward** To prevent reward sparsity during early training, we measure semantic relevance between $c$ and the ground-truth caption $c^\star$ using ROUGE and BLEU-4, as shown in Eq. 5:

$$R_{\text{caption}} = w_r \cdot ROUGE(c, c^\star) + (1 - w_r) \cdot BLEU(c, c^\star) \tag{5}$$

where $w_r$ represent the weight of ROUGE score, set as 0.7 in the experiments.

Figure 5: **The reward function**. Given the Generated caption (G), Question (Q), and Answer (A), the reward function measures the caption's quality from two aspects: 1) its reasoning reward, and 2) caption reward, as forumated as $R(c, I) = \lambda_r \cdot R_{\text{reasoning}}(c, q) + (1 - \lambda_r) \cdot R_{\text{caption}}(c, c^\star)$. **Reasoning reward** stands for the caption's relevance to the image and question, especially the ability to capture the key reasoning information for solving the question. **Caption reward** is an auxiliary reward signal that measures the caption's similarity to the ground truth caption.

## 4 EXPERIMENTS

### 4.1 EXPERIMENTAL SETUP

Our experiments utilize Gemma3-4B (Farabet & Warkentin, 2025), a commonly used lightweight multimodal architecture with strong mathematical reasoning capabilities, as our base model. All in-domain experiments are conducted on MathVista (Lu et al., 2023) and MathVerse (Zhang et al., 2024a), two established mathematical reasoning benchmarks focusing on visual and mathematical problem-solving.

### 4.2 IN-DOMAIN PERFORMANCE OF GEOREASONING-10K

In this section, we verify the effectiveness and scalability of our proposed dataset on in-domain benchmarks. Several commonly-used datasets in this field are chosen as baselines, including Auto-Geo (Huang et al., 2025b), GeoPeP (Sun et al., 2025), GeoGPT4V (Cai et al., 2024), Geo170K (Gao et al., 2023), GeoQA (Chen et al., 2021), and MathVision Wang et al. (2024).

It can be observed from Table 1 that the model trained on GeoReasoning-10K obtains better mathematical reasoning performance compared to that trained on other caption datasets. This improvement mainly concentrates on in-domain mathematical subtasks, such as geometry, algebra, science, statistics, and most subtasks in MathVerse. The performance gain can be attributed to the symbolic synthesis process of our pipeline, which allows an infinite number of possible geometry problem types and offers diverse difficulty levels for the generated images.

Besides, the performance of models trained on the full datasets on MathVista and MathVerse is shown in Table 4 in Appendix D.

Table 1: **Better In-Domain Performance**. Accuracy of Gemma3-4B models trained on 10k random samples of each dataset over 4 trials. Results on several subtasks in MathVerse and MathVista are also reported here.

| | MathVista (↑) | | | | | MathVerse (↑) | | | |
|---|---|---|---|---|---|---|---|---|---|
| | Overall | Geometry | Algebra | Science | Statistic | Overall | Vision -Dominant | Text -Dominant | Text -Lite |
| Base | 46.2 | 60.7 | 59.1 | 53.3 | 43.2 | 25.2 | 24.0 | 32.0 | 25.9 |
| AutoGeo | 47.8±0.8 | 62.3±2.4 | 60.2±1.9 | 52.5±1.2 | 44.1±0.9 | 24.6±0.4 | 22.3±1.4 | 35.2±0.7 | 26.7±1.3 |
| GeoPeP | 47.5±0.4 | 61.0±2.3 | 59.6±1.8 | 54.1±0.6 | 44.2±0.8 | 24.2±0.2 | 21.7±0.9 | 33.7±0.3 | 25.7±1.3 |
| GeoGPT4V | 47.5±0.2 | 60.5±0.7 | 59.3±1.3 | 54.1±1.5 | 44.6±1.0 | 25.2±0.5 | 22.4±0.8 | 36.4±1.4 | 26.9±1.0 |
| Geo170K | 47.6±0.3 | 62.2±1.5 | 60.6±1.2 | 53.5±1.5 | 43.7±0.4 | 25.3±0.1 | 22.5±1.0 | 35.4±1.7 | 26.9±0.7 |
| GeoReasoning | **48.6±0.3** | **62.8±1.3** | **61.4±1.4** | **54.3±1.2** | **46.0±0.5** | **25.8±0.1** | **24.0±0.8** | **36.8±0.4** | **28.4±0.5** |

GeoReasoning also demonstrates better scalability, as shown in Figure 6. The model trained on GeoReasoning improves progressively when the dataset sizes increase. Moreover, GeoReasoning

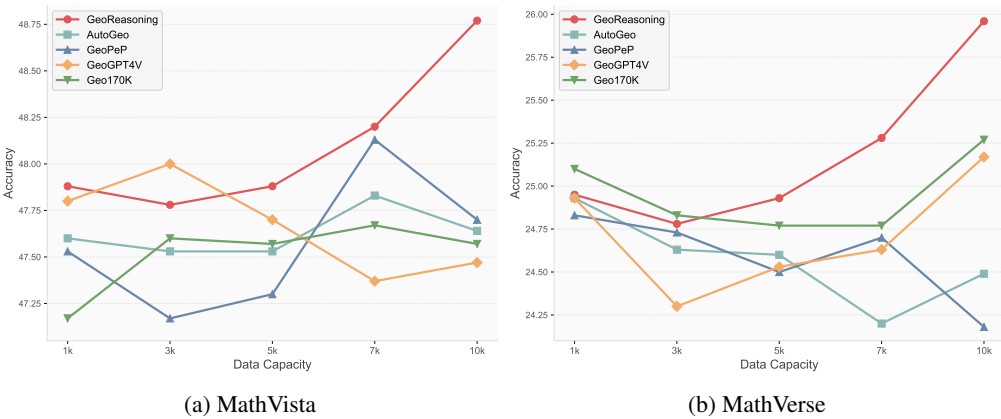

(a) MathVista
(b) MathVerse

Figure 6: **Better Scalability**. The accuracy of models fine-tuned on different capacities and mathematical datasets on downstream evaluation benchmarks: (a) MathVista. (b) MathVerse.

outperforms existing datasets by a non-trivial margin at 10K size, verifying the effectiveness of the proposed cross-modal alignment method. In this setting, subsets of different sizes are randomly sampled from the original dataset, with the baseline models trained on these subsets.

### 4.3 OUT-OF-DOMAIN PERFORMANCE OF GEOREASONING-10K

Surprisingly, GeoReasoning-10K also demonstrates better out-of-domain generalization ability for non-geometric input images. Specifically, we evaluate the accuracy of the baseline models and the trained models on a commonly-used benchmark MMMU (Yue et al., 2023), as shown in Table 2.

Table 2: **Better Out-of-Domain Performance**. Accuracy of models evaluated on all subtasks of MMMU over 5 trials, where "A&D.", "Busi.", "Sci.", "H&M.", "Human.", "Tech." are short for "Art and Design", "Business", "Science", "Health and Medicine", "Humanities and Social Science", "Tech and Engineering", respectively.

|  | Overall | A&D. | Busi. | Sci. | H&M. | Human. | Tech. |
|---|---|---|---|---|---|---|---|
| Base | 43.3±0.7 | 57.8±4.0 | 44.1±0.6 | 34.3±0.9 | 46.8±2.2 | 59.2±2.1 | 29.0±1.3 |
| AutoGeo | 43.5±0.5 | 59.3±1.4 | 43.3±1.1 | 34.9±1.3 | 47.4±1.1 | 58.9±1.5 | 30.7±2.6 |
| GeoPeP | 43.7±0.9 | 59.2±1.1 | 40.4±1.4 | 34.0±1.7 | 45.1±0.9 | 59.6±1.0 | 32.6±0.7 |
| GeoGPT4V | 44.0±0.7 | **60.2±1.1** | 43.1±1.5 | 34.5±0.7 | 46.0±0.7 | 58.3±1.6 | 30.8±2.0 |
| Geo170K | 42.9±1.0 | 58.5±0.8 | 43.6±1.4 | 30.9±2.0 | 46.8±2.2 | 59.9±2.2 | 30.9±1.6 |
| GeoReasoning | **44.9±0.7** | **60.2±2.0** | **44.5±2.5** | **36.0±2.0** | 46.7±1.1 | **60.0±0.5** | **32.9±1.3** |

It can be observed from Table 2 that the trained model outperforms the baseline on most of the domains, especially subtasks involving line art or drawing, such as Art & Design, and Tech & Engineering. These types of improvement can be mainly attributed to the proposed RLVR training process, whose reasoning reward forces the model to focus on the key element in the geometric image for problem-solving. This key-element capturing ability can help model generalize better to scenarios other than geometric problems.

We also conduct qualitative analyses on representative examples from MathVista and MMMU to understand which types of reasoning abilities our method generalizes especially well. As shown in Figure 1, 11, and 12, GeoReasoning is in particular favorable for arithematic reasoning, line-art reasoning, and spatial reasoning. Additional examples are also available in Appendix C.

### 4.4 ABLATION STUDY

To understand the individual contribution of Cold-Start and RLVR phases, additional ablation studies are conducted. Specifically, we implement the Cold Start and RLVR pipelines on Gemma3-4B, generating refined models and datasets at each optimization stage, as shown in Table 3.

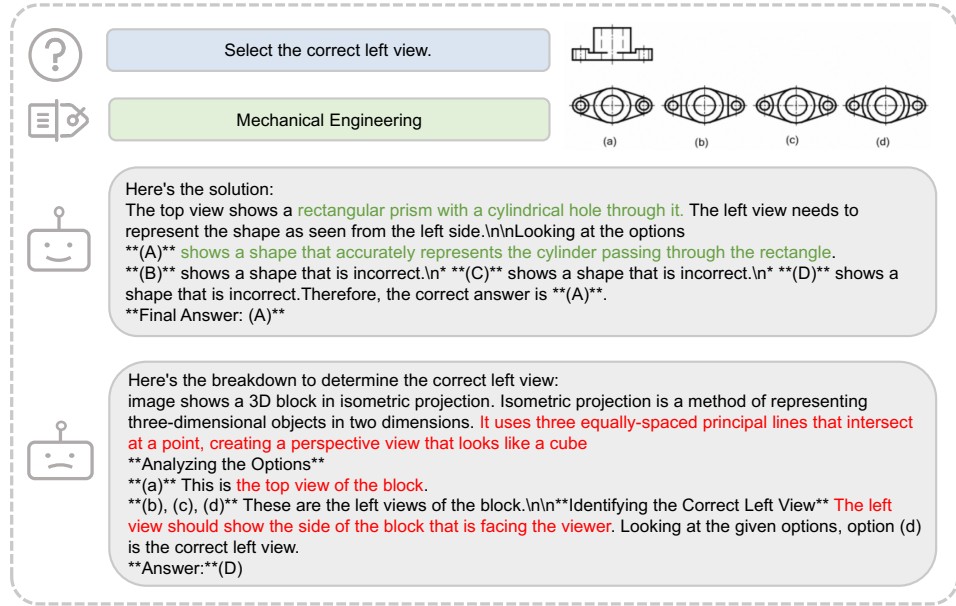

Figure 7: **Cast Study**. An engineering case, where the base model's answer is relatively general and the analysis of shape is not rigorous enough, while the model after training on GeoReasoning is more detailed and accurate in observing shape and has spatial reasoning ability.

Table 3: **Ablation Study.** Accuracy of Gemma3-4B models with various stages.

| Stage 1: Cold Start | Stage 2: RAFT | MathVerse | MathVista |
|---|---|---|---|
| ✗ | ✗ | 25.2 | 46.2 |
| ✓ | ✗ | 25.9 (+0.7) | 47.6 (+1.4) |
| ✗ | ✓ | 26.1 (+0.9) | 49.4 (+3.2) |
| ✓ | ✓ | **27.4 (+2.2)** | **50.0 (+3.8)** |

It can be observed that RLVR benefits both the base model and the one after cold start, demonstrating that accuracy-guided captioning enhances the model's general geometric problem-solving ability on in-domain questions.

When it comes to out-of-domain generalization, such as most non-geometric samples in MathVista, RAFT plays a more important role, contributing most of the improvements. This implies that the RLVR stage enables the captioner to focus more on key aspects of the geometric image for problem-solving, resulting in higher-quality captions for generalization. For example, captions like "the line's color is blue" or "the left number is 26" are mostly irrelevant, whereas a caption like "the length of line AB equals line BC" is more useful for generalization purposes.

## 5 CONCLUSION

In this paper, we propose **Geo-Image-Textualization**, a novel reinforcement learning-based framework designed to symbolically synthesize high-quality, geometry-centered multimodal data. Leveraging this framework, we construct **GeoReasoning-10K**, a new dataset aimed at bridging the gap between visual and linguistic modalities in the geometry domain. Extensive experiments on the MathVista and MathVerse benchmarks demonstrate that our dataset significantly enhances the cross-modal reasoning capabilities of trained MLLMs, with improvements generalizing to non-geometric domains. Detailed analysis shows that geometric image caption datasets are especially beneficial for generalized skills of arithmetic reasoning, line-art reasoning, and spatial reasoning.

## ETHICS STATEMENT

After carefully reviewing the ethical regulations of the conference, to the best of our knowledge, this work does not present any foreseeable ethical concerns. No negative societal or ethical impacts are anticipated for the contribution of this work. The proposed GeoReasoning-10K dataset only contains geometric images and corresponding captions, does not involve anything about human subjects, potentially harmful insights, potential conflicts of interest and sponsorship, discrimination/bias/fairness concerns, privacy and security issues, legal compliance, and research integrity issues.

## REPRODUCIBILITY STATEMENT

We have made efforts to ensure that our work is reproducible, with details provided in Section 4 and Appendix A, B. The code for data generation and RLVR will be released upon the acceptance of the paper.

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

## A    QUESTION-ANSWER PAIR GENERATION PROMPT

The rule-and-LLM-based pipeline contains two stages. We first design a prompt that satisfies all the above conditions, using a relatively low temperature (0.2 in our experiments) to encourage the large model (Gemini 2.5 Flash) to generate initial questions based on the caption, while also labeling those that are inconsistent with the caption. For the inconsistent questions, we then switch to a different prompt, encouraging the model to incorporate additional information and formulate new questions accordingly, while increasing the temperature to 0.8. This process continues until a self-consistent question is generated for the first time.

The prompt of the first question generation stage is set as:

---

**Prompt1**

```
You are a helpful dataset processor.  Please:
1.  Generate a mathemetical question according to the
given description of a geometric image with the following
requirements:
1.1 The problem should base on the given description, i.e., you
must **NOT** generate problems that are unrelated to the given
description.
1.2 The problem difficulty should not be too low, such as
determining some information in the description.
1.3 It should also not be too hard, like introducing too much
extra information, but anyway you can introduce some extra
information to form a good geometric problem.
1.4 You should **NOT** include or repeat any information in the
description, and just contain the real question.  For example,
if the description says:  'Line segment AB is present.  The
length of BA is 1.24.', then when you generate the question,
you should not include the length of AB.
1.5 If the question is inconsistent with the given description,
the final answer should be 'None'.
2.  Answer the question you just provided, and express the
final answer to two decimal places.  The final answer should
be in \\boxed{{answer}}.

Description:
{description}
Generated Question:
{question}
Generated Response:
{response}
Final Answer:
\\boxed{{answer}}
```

---

The prompt of the question re-generation stage is set as:

---

**Prompt1**

```
You are a helpful dataset processor.  Please:  1.  Generate a
mathemetical question according to the given description of a
geometric image with the following requirements:
1.1 The problem should base on the given description, i.e., you
must **NOT** generate problems that are unrelated to the given
description.
1.2 You can introduce some extra information to form a good
geometric problem.
1.3 If you find that it is hard to generate some difficult
questions, just give a simple question.  For example, when the
lengths of all four sides of a quadrilateral are given, you can
no longer assume it is a parallelogram or rectangle.  In such
cases, the problem may only allow for questions like asking for
the perimeter, or determining the length of a segment when a
certain point divides a side into an n-equal part, etc.
1.4 You should **NOT** include or repeat any information in the
description, and just contain the real question.  For example,
if the description says:  'Line segment AB is present.  The
length of BA is 1.24.', then when you generate the question,
you should not include the length of AB.
1.5 If the question is inconsistent with the given description,
the final answer should be 'None'.
2.  Answer the question you just provided, and express the
final answer to two decimal places.  The final answer should
be in \\boxed{{answer}}.
Description:
{description}
Generated Question:
{question}
Generated Response:
{response}
Final Answer:
\\boxed{{answer}}
```

---

## B  EXPERIMENTAL DETAILS

### B.1  EXPERIMENTAL SETUP

The training and optimization pipeline contains two stages:

1. **Cold-Start phase**: we train each base model on the initial GeoReasoning-10K dataset for 1 epoch using standard supervised fine-tuning (SFT). The peak learning rate is $10^{-5}$, with a cosine learning rate scheduler and a 0.03 warm-up ratio for linear warm-up.

2. **RLVR phase**: we run RAFT (Dong et al., 2023) for 5 epochs, alternating between two sub-phases:
   - 2.1) Caption Refinement: The model generates 8 candidate captions for each image, and the top-1 caption per image is selected based on a composite reward with reasoning reward weight of $\lambda_r = 0.7$ and caption reward weight of $1 - \lambda_r = 0.3$.
   - 2.2) Model Retraining: Fine-tune the model on the selected dataset for 1 epoch using the same hyperparameters as the cold start phase.

To ensure consistency, we adopt the official evaluation codebases of both MathVerse and MathVista, using the GPT-4o-mini API to evaluate the performance of our MLLM. Specifically, following each benchmark's official setup, we use GPT-4o-mini to extract and assess the correctness of answers for MathVerse, and to extract answers for MathVista.

We evaluate MLLMs on MathVerse, MathVista and MMMU using A100 by VLLM. We employ Gemma3-4B as our base model and fine-tune it on Georeasoning-10K using 4 L20 GPUs. The training process is distributed using torchrun with the DeepSpeed ZeRO-3 optimization strategy.

### B.2  DATA SOURCE

GeoReasoning-10K dataset is generated through rule-based methods and further refined using the RAFT framework. The question-answer pairs are generated by Gemini 2.5-Flash with a specific prompt.

### B.3  LICENSE

- **GeoReasoning-10K** is released under the MIT License.
- **MathVerse** and **MathVista** are evaluated using their official codebases, which are publicly available under the Apache 2.0 License and MIT License, respectively.
- Our use of the **GPT-4o-mini API** for evaluation complies with OpenAI's API usage policies.
- All third-party datasets and models used in this work are under their respective licenses, and we ensure compliance with their terms of use.

## C  CASE STUDIES

Additional mathematical cases that further demonstrate the generalization capability of our RAFT method are presented here. Figure 8, Figure 9, and Figure 10 correspond to the geometric, arithmetic, and numeric domains, respectively.

Figure 11 and Figure 12 exhibit non-mathematical cases of the economics and physics domains, respectively.

All these examples indicate that training on geometric caption tasks stimulates the reasoning capacity of models.

## D  COMPARISON TO OTHER DATASETS

We test the model performance trained on total sizes of AutoGeo, GeoPeP, GeoGPT4V, Geo170K, GeoQA, MathVision, with the accuracy of MathVista and MathVerse shown in Table 4:

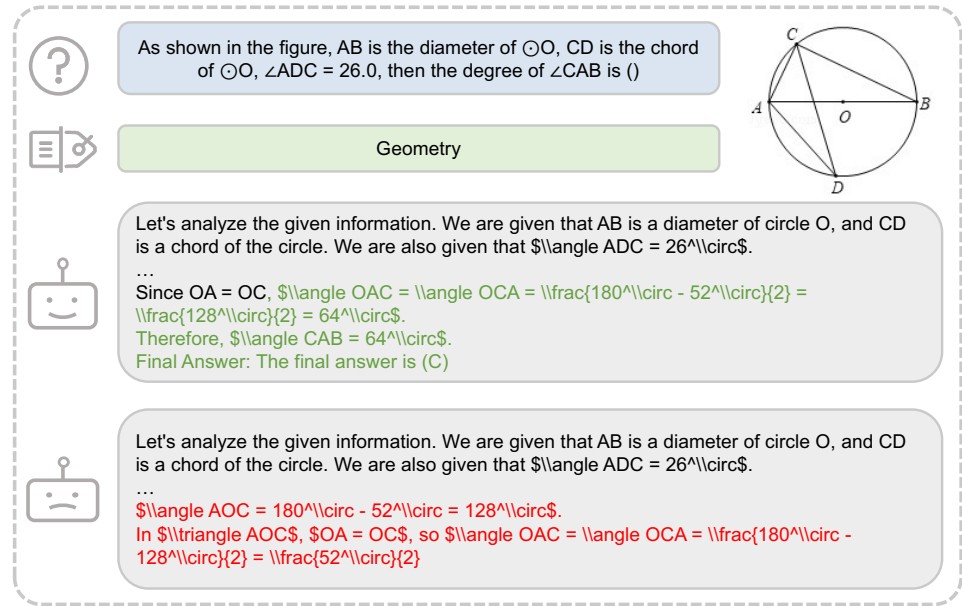

Figure 8: A geometric case.

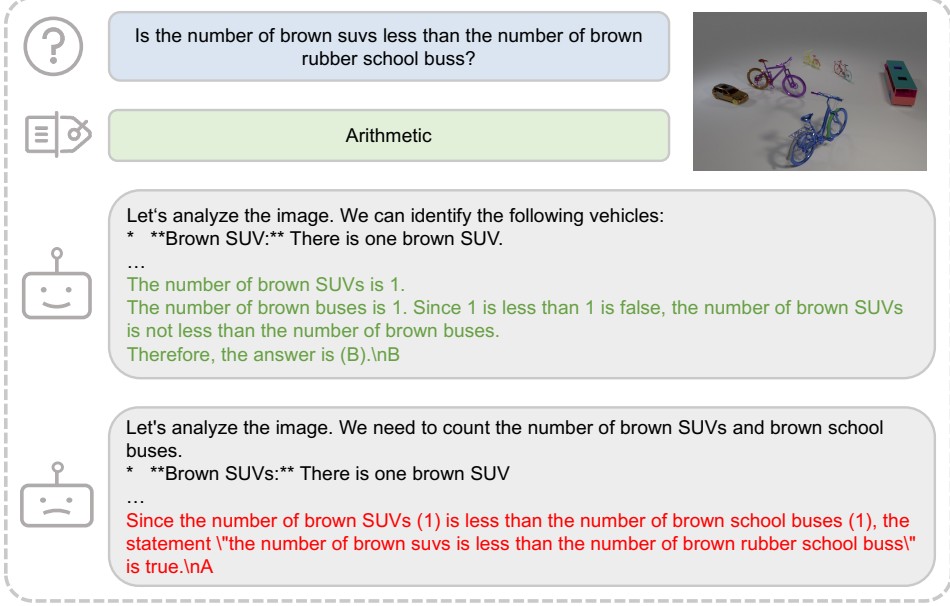

Figure 9: An arithmetic case.

## E  ADDITIONAL ABLATION STUDIES

This section serves as a complement of Section 4.4, exhibit ing ablation studies on various domains, and hyperparameters of the reward function.

### E.1  ABLATION STUDY ON VARIOUS DOMAINS

We record the skills across diverse domains like geometry and arithmetic on various training stages in Table 5 and Table 6:

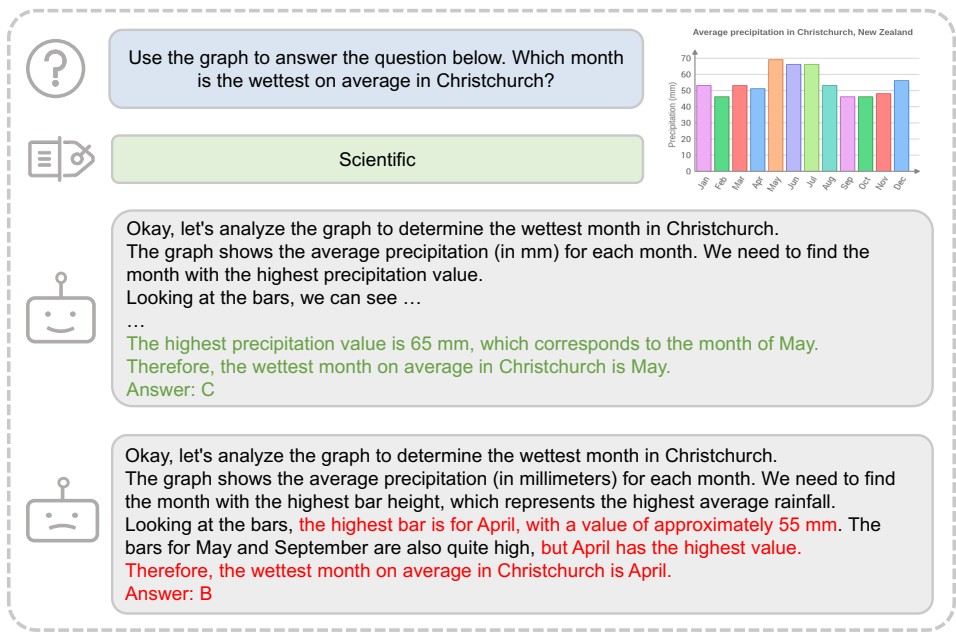

Figure 10: A numeric case.

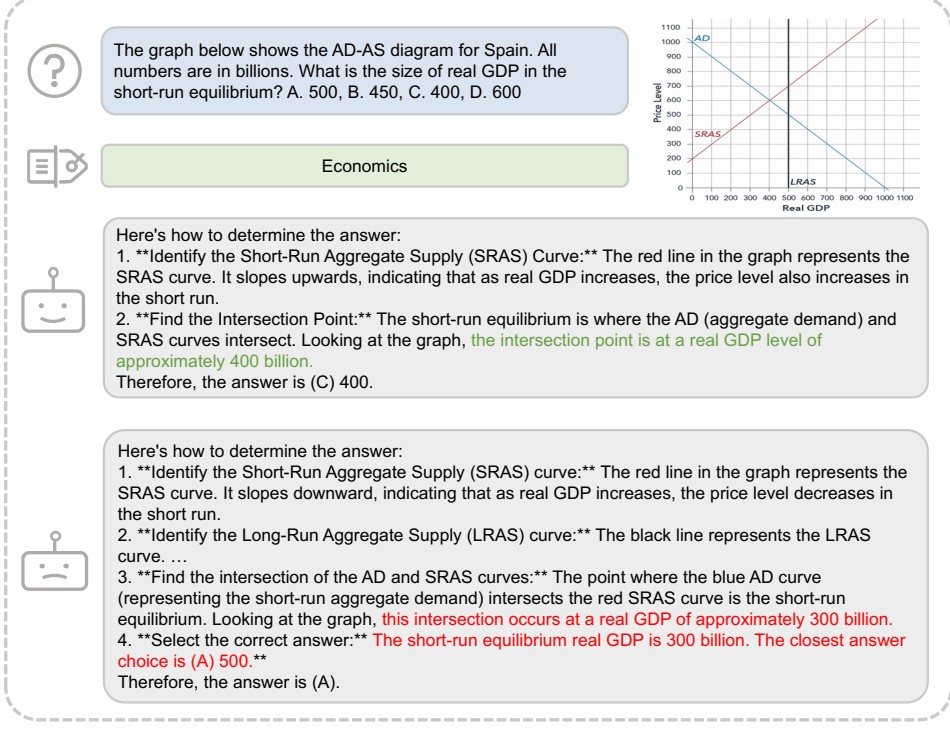

Figure 11: An economics case.

It can be observed from Table 5 and Table 6 that the model after RAFT stages outperforms the base model across all domains. Specifically, the model achieves significant performance improvements across the arithmetic, algebraic, and numeric domains, with respective gains of 5.1%, 4.2%, and 5.5%. These results demonstrate the effectiveness of our approach in enhancing model performance as well as its generalization capability across different domains.

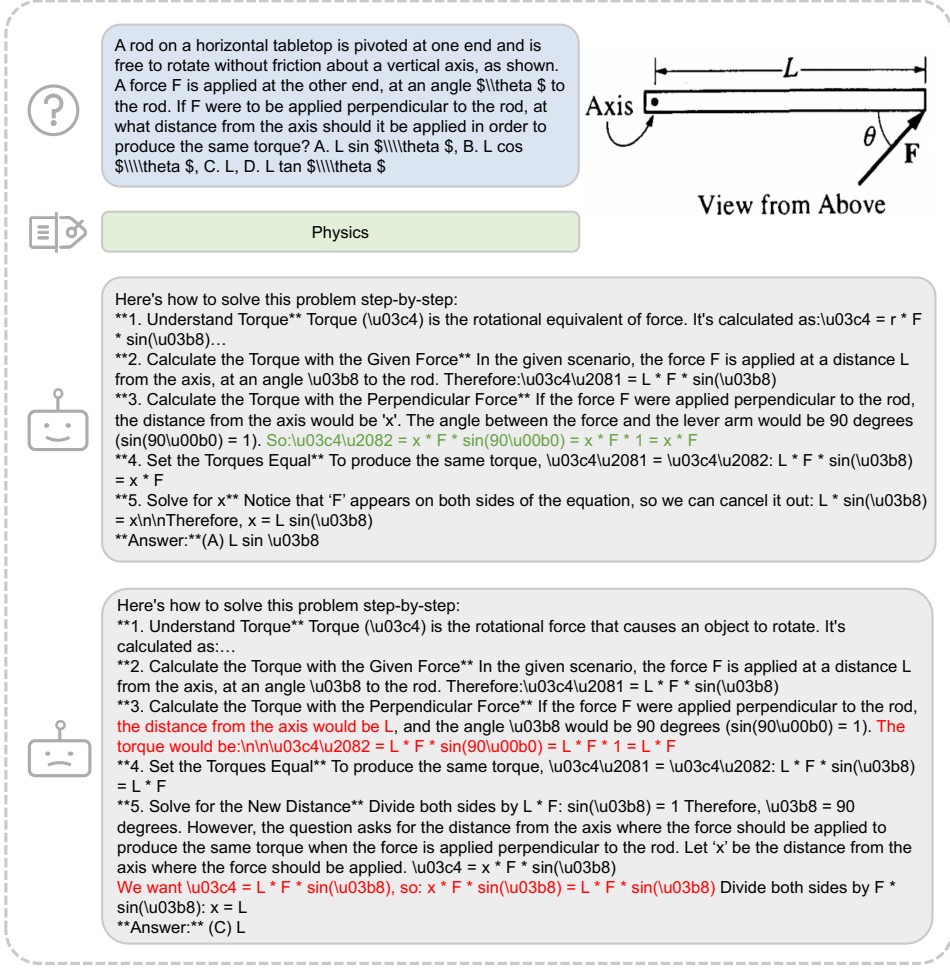

Figure 12: A physics case.

Table 4: Performance of Gemma3-4B models trained on the total capacity of our dataset and counterpart datasets.

|  | Capacity | MathVerse | MathVista |
|---|---|---|---|
| Geo170K | 117k | 22.0 | 46.8 |
| GeoPeP | 100k | 22.7 | 47.1 |
| GeoGPT4V | 23k | 24.6 | 46.0 |
| AutoGeo | 100k | 24.7 | 46.1 |
| MathVision | 3k | 24.7 | 46.9 |
| GeoQA | 5k | 24.9 | 46.0 |
| GeoReasoning | 10k | **25.8** | **48.6** |

## E.2 ABLATION STUDY ON HYPERPARAMETERS

We evaluated the RAFTed models with various hyperparameters on MathVista and MathVerse, as shown in Table 7:

As shown in Table 7, the reasoning reward plays a more important role in MathVista than MathVerse, indicating that the gain of genelization comes more from the helpness in solving the question other than comparison with captions.

Table 5: Accuracy of Gemma3-4B models at various stages tested on MathVista

|  | baseline | cold-start | raft-1 | raft-2 | raft-3 | raft-4 | raft-5 |
|---|---|---|---|---|---|---|---|
| all | 46.2 | 47.6 | 48.7 | 48.1 | 49.2 | 49.0 | **50.0** |
| geometry | 60.7 | 62.3 | 63.2 | 64.0 | 63.6 | 60.3 | **64.0** |
| arithmetic | 42.5 | 45.0 | 44.8 | 45.3 | 45.9 | **47.6** | 46.5 |
| algebraic | 59.1 | 60.5 | 62.3 | 62.3 | 62.3 | 59.1 | **63.3** |
| numeric | 26.4 | 31.9 | 29.9 | 31.3 | 31.3 | 31.9 | **31.9** |

Table 6: Accuracy of Gemma3-4B models at various stages tested on MathVerse

|  | baseline | cold-start | raft-1 | raft-2 | raft-3 | raft-4 | raft-5 |
|---|---|---|---|---|---|---|---|
| all | 25.2 | 25.9 | 25.7 | 25.8 | 25.5 | 26.5 | **27.4** |
| text dominant | 32.0 | 35.5 | 35.1 | 35.2 | 35.1 | 36.5 | **36.5** |
| text lite | 25.9 | 27.4 | 28.2 | **28.5** | 27.4 | 26.6 | 26.3 |
| vision intensive | 24.0 | 24.8 | 24.4 | 24.4 | 23.1 | 26.1 | **26.5** |

Table 7: Accuracy of RAFTed models with various hyperparameters evaluated on MathVista and MathVerse, where $\lambda_r$ stands for the weight of reasoning reward.

|  | MathVista | MathVerse |
|---|---|---|
| $\lambda_r = 1$ | 49.8 | **27.5** |
| $\lambda_r = 0.7$ | **50.0** | 27.4 |
| $\lambda_r = 0$ | 48.9 | 27.5 |

In addition, it is observed in the result that the performance is not very sensitive to the selection of this hyperparameter, indicating the robustness of our RAFT method.

## F  THE SCALING OF GEOREASONING

To further understand the scaling property of GeoReasoning, we scale the dataset to 20K samples. We train the base model on this expanded dataset using exactly the same hyperparameters as in the prior experiments, and then validate its performance on MathVista and MathVerse.

As shown in Figure 13, the generalization performance keeps improving from 10K to 20K. This property is different from AutoGeo and GeoPeP (Table 1 and Table 4), which show performance degradation when scaling up the training samples, demonstrating the higher quality of GeoReasoning compared to them.

Besides, we evaluate the out-of-domain performance on MMMU, as shown in Table 8.

Table 8: **Scaling from 10K to 20K**. The performance of models trained on GeoReasoning with 10K or 20K samples on MMMU.

|  | MMMU |
|---|---|
| GeoReasoning-10K | 44.9±0.7 |
| GeoReasoning-20K | **45.2±0.5** |

As shown in Table 8, the out-of-domain performance improves from 10K to 20K, indicating the strong power of GeoReasoning to improve the generalization capacity of models.

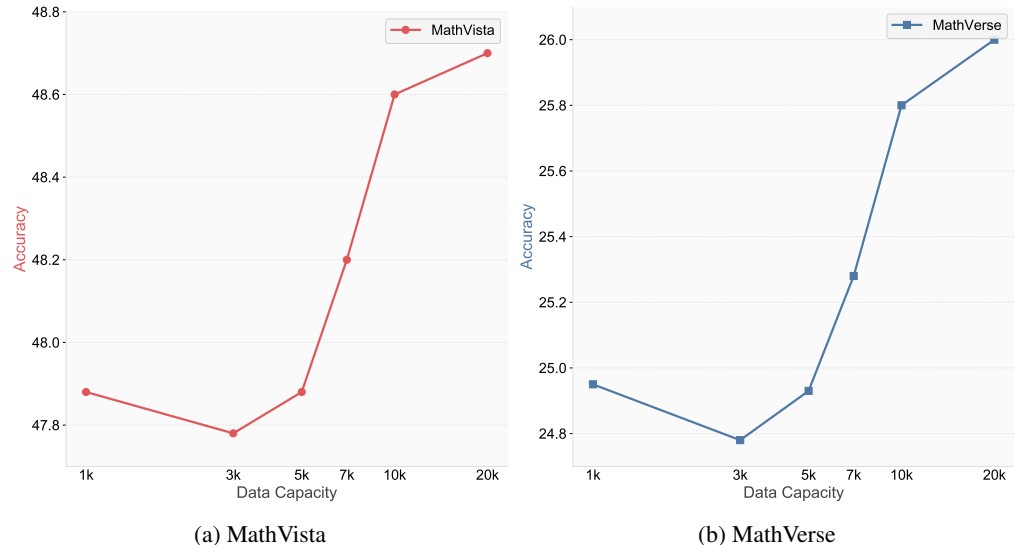

(a) MathVista          (b) MathVerse

Figure 13: **Scalability with more samples**. The accuracy of models fine-tuned on different capacities and mathematical datasets on downstream evaluation benchmarks: (a) MathVista. (b) MathVerse.

## G   THE USE OF LARGE LANGUAGE MODELS

ChatGPT and GPT-5 were adopted to polish the writing of the paper, where all revised sentences were double-checked by the authors. ChatGPT was also utilized to write external parallelization scripts to speed up the image generation process, and the scripts were carefully reviewed by the authors.

## H   BROADER IMPACTS

The provided dataset pipeline and the generated dataset contribute to enhancing the generalizable reasoning abilities of multimodal large language models (MLLMs). In narrow domains, they are particularly effective for improving the geometric problem-solving capabilities of MLLMs, while in broader domains, they support the development of mathematical reasoning skills applicable to everyday scenarios. As the dataset is limited to geometric mathematical problems, it is considered safe for release and is unlikely to pose direct negative social impacts.

