# OpenReview forum: "Generalizable Geometric Image Caption Synthesis"
_ICLR.cc/2026/Conference — Submitted to ICLR 2026_

### Official Review · Reviewer_hdrJ · 2025-10-28

**Soundness:** 2
**Presentation:** 2
**Contribution:** 2
**Rating:** 2
**Confidence:** 5

**Summary:**

This paper introduces GeoReasoning-10K, a dataset of 10,000 high-quality image-caption pairs designed to enhance the cross-modal reasoning capabilities of multimodal large language models (MLLMs) in geometric problem-solving. The authors propose a novel Geo-Image-Textualization framework, integrating Reinforcement Learning with Verifiable Rewards (RLVR) to iteratively refine data quality and model performance. Experimental results demonstrate significant improvements in both in-domain tasks (e.g., geometry, algebra) and out-of-domain generalization (e.g., art, design).

**Strengths:**

1.	The paper proposes the Geo-Image-Textualization framework, which combines reinforcement learning and reward mechanisms (RLVR) to significantly improve the semantic alignment between geometric images and textual descriptions.
2.	Comprehensive experiments across 3 benchmarks (MathVista, MathVerse, MMMU) with multiple trials and error bars

**Weaknesses:**

1.	Figure 1 is unconvincing. How can training on geometry enable the model to calculate "the age difference between King Richard III and Queen Anne Neville"? This generalization lacks theoretical support. A more plausible explanation is that increased model scale (more training) improves general capabilities, rather than transfer from geometric reasoning. Additionally, Figure 1 is poorly placed—not referenced until Section 4.3.
2.	The method description needs more details: how are the geometric relationships constructed? Does it cover all core concepts of a standard geometry curriculum? What are the RELATION SAMPLING rules?
3.	Explicitly encoding semantic relationships within the image is not surprising, as existing datasets (e.g., MAVIS, TR-COT) achieve similar effects. It's unclear how its quality compares to them. Additionally, the paper lacks discussion on how this annotation impacts generalization.
4.	The improvement in experimental results in Table 1 is minimal, with significant overlap in standard deviation.
5.	The citation of Figure 5 in Section 3.3 seems to be incorrect.

**Questions:**

See the weaknesses

---

> ### Author Response · Authors · 2025-11-27
> **R1: Response to Reviewer hdrJ - Part 1**
>
> Thank you so much for your valuable suggestions and insightful comments. We deeply appreciate your recognition of the significance and comprehensiveness of experiments in our work. We would like to address your concerns as follows:
>
> ### Weakness 1: Clarification on Figure 1
> > Figure 1 is unconvincing. How can training on geometry enable the model to calculate "the age difference between King Richard III and Queen Anne Neville"? This generalization lacks theoretical support. A more plausible explanation is that increased model scale (more training) improves general capabilities, rather than transfer from geometric reasoning. Additionally, Figure 1 is poorly placed—not referenced until Section 4.3.
>
> Thanks for the question. This is because, in GeoReasoning **the arithmetic calculation abilities are improved**. For example, in Table 1, its performance in Algebra is improved by a nontrivial margin.
>
> |              | MathVista (Algebra) |
> | ------------ | ------------------- |
> | Base         | 59.1%               |
> | Geo170K      | 60.2%               |
> | GeoPep       | 59.6%               |
> | GeoGPT4V     | 59.3%               |
> | Geo170K      | 60.6%               |
> | GeoReasoning | **61.4%**           |
>
> And **the improvement is not caused by more training**. Table 4 in the appendix provides evidence that answers this question. In Table 4, it is shown that the model with more training actually underperforms GeoReasoning, which demonstrates GeoReasoning's better generalization performance.
>
> |                    | MathVerse | MathVista |
> | ------------------ | --------- | --------- |
> | Geo170K            | 22.0%     | 46.8%     |
> | GeoPep (100K)      | 22.7%     | 47.1%     |
> | GeoGPT4V (23K)     | 24.6%     | 46.0%     |
> | AutoGeo (100K)     | 24.7%     | 46.1%     |
> | GeoReasoning (10K) | **25.8%** | **48.6%** |
>
> Regarding Figure 1, thanks for the helpful suggestion. We have added a reference to the figure in the introduction accordingly to highlight this point. Thanks for pointing that out!
>
>
> ### Weakness 2: Clarification on Data Generation Pipeline
> > The method description needs more details: how are the geometric relationships constructed? Does it cover all core concepts of a standard geometry curriculum? What are the RELATION SAMPLING rules?
>
> Thanks for the question! The data generation pipeline is detailed in Figure 3 and its caption, with **geometric relationships sampled from a fixed library of 50 types**, e.g., drawing a circle centered at a point, bisecting an angle, creating an equilateral triangle, etc.
>
> To ensure its coverage, we randomly sampled 100 images from MathVerse using its official visualization tool. **All the relations of the sampled images in MathVerse can be generated by our pipeline**, showing that the coverage of the employed relation library is sufficient.
>
> Regarding the relation sampling rules, we perform random sampling from the 50 types each time. If the sampled relationship conflicts with the existing relations, the random sampling process is repeated until a valid relation is found. We will include this detail in the caption of Figure 3 in our revisions as well.
>
> ### Weakness 3: More Baselines
> > Explicitly encoding semantic relationships within the image is not surprising, as existing datasets (e.g., MAVIS, TR-COT) achieve similar effects. It's unclear how its quality compares to them. Additionally, the paper lacks discussion on how this annotation impacts generalization.
>
> Thanks for the constructive comments. To address this concern, we have conducted additional experiments to include the mentioned datasets, following the same setting in the paper, respectively, as shown in the following table:
>
> |  | MathVerse | MathVista |
> | - | - | - |
> | MAVIS-10K (random sampled) | 25.0% | 45.9% |
> | GeoMM-10K (random sampled) | 24.9% | 47.7% |
> | MAVIS-200K | 24.2% | 48.1% |
> | GeoMM-100K | 25.1% | **48.6%** |
> | GeoReasoning (10K) | **25.8%** | **48.6%** |
>
> As shown in the table, GeoReasoning still matches or outperforms both baselines even with a significantly smaller number of samples.

---

> > ### Author Response · Authors · 2025-11-27
> > **R1: Response to Reviewer hdrJ - Part 2**
> >
> > ### Weakness 4: Clarification on Experimental Results
> > > The improvement in experimental results in Table 1 is minimal, with significant overlap in standard deviation.
> >
> > Thanks for the comment. The comment is not true for the most salient Overall metric, where GeoReasoning provides +2.4%/+0.6% accuracy improvements with only $\pm$0.3%/0.1% standard deviation. Even compared to other baseline methods, GeoReasoning's improvement is at least one standard deviation away. This also holds for the subtasks of Statistics (MathVista) and Text-Lite (MathVerse), where GeoReasoning significantly outperforms all the baselines.
> >
> > ### Weakness 5: Minor: Incorrect Image Label
> > > The citation of Figure 5 in Section 3.3 seems to be incorrect.
> >
> > Thanks for pointing that out! We have modified it in the latest revision accordingly.
> >
> >
> > If you believe our responses have satisfactorily addressed your concerns, we would be most appreciative if you could consider raising the score.
> >
> > We fully understand the demands on your time and sincerely thank you for your efforts in helping us improve our work. We look forward to any additional feedback you may have.

---

> ### Author Response · Authors · 2025-11-28
> **Thank you for updating the score**
>
> Dear Reviewer hdrJ,
>
> Thank you very much for the positive feedback. We understand that the editing button for score revising is missing (maybe due to a bug), and is causing some trouble on the reviewers' end.
>
> Meanwhile, please feel free to let us know if there are any remaining concerns or additional revisions needed. We are eager to further enhance our manuscript during the rebuttal period and would greatly appreciate any additional guidance from you.
>
> Thank you again for your valuable time and assistance!

---

### Official Review · Reviewer_183W · 2025-10-30

**Soundness:** 3
**Presentation:** 3
**Contribution:** 2
**Rating:** 6
**Confidence:** 3

**Summary:**

This paper addresses the poor geometric reasoning and cross-modal generalization capabilities of multimodal large language models (MLLMs), stemming from the lack of high-quality, fully aligned geometry image-text datasets and the poor generalization of existing symbolic synthesis pipelines. To solve these issues, the authors propose a reinforcement learning-based framework called Geo-Image-Textualization and construct a dataset named GeoReasoning-10K (10,000 fully aligned geometry image-caption pairs). The framework integrates a Reinforcement Learning with Verifiable Rewards (RLVR) phase, which iteratively optimizes dataset quality and model performance through alternating caption refinement and model retraining. Experimental results show that MLLMs trained on GeoReasoning-10K achieve significant accuracy improvements on in-domain benchmarks (MathVista, MathVerse) and out-of-domain tasks (MMMU’s Art & Design, Tech & Engineering).

**Strengths:**

1. The paper demonstrates strong originality through targeted innovation in addressing unmet needs and creative fusion of existing methods. First, it identifies a critical gap in multimodal large language model (MLLM) research—poor geometric reasoning due to misaligned image-text data—an underexplored niche despite MLLMs’ success in general tasks.
2. The RLVR (Reinforcement Learning with Verifiable Rewards) Phase reimagines reinforcement learning (RL) for dataset refinement: instead of using RL to optimize model inference directly, it creates a "data-model co-evolution loop"—a creative combination of RL, captioning, and reasoning evaluation.
3. Experimentally, the authors use rigorous ablation studies (e.g., testing with/without RLVR, with/without cold-start) to isolate the impact of each component, avoiding conflation of results. They validate on multiple benchmarks (MathVista, MathVerse, MMMU) across in-domain and out-of-domain tasks, ensuring conclusions are not benchmark-specific.

**Weaknesses:**

1. The paper presents GeoReasoning-10K as a "high-quality aligned dataset," yet it offers insufficient detail about how well the dataset covers complex geometric edge cases—coverage that is essential for evaluating true model generalization. In contrast, earlier geometric reasoning datasets such as GeoQA explicitly document their diversity in edge cases, whereas GeoReasoning-10K lacks similar transparency. To address this, the authors should publish a detailed breakdown of the dataset, categorizing samples by geometric type (e.g., 2D vs. 3D, Euclidean vs. non-Euclidean), complexity (e.g., number of components, number of inference steps required), and other relevant dimensions.
2. The core innovation of the RLVR phase is the composite reward function $R = \lambda_r \cdot R_{\text{reasoning}} + (1 - \lambda_r) \cdot R_{\text{caption}}$ (with $\lambda_r = 0.7$), but the paper fails to validate whether both components are truly necessary—for example, by testing variants that use only one of the two rewards. Moreover, the choice of $\lambda_r = 0.7$ appears arbitrary, as the paper provides no justification or analysis for this hyperparameter setting.
3. The RLVR Phase involves 5 rounds of "caption generation → reward scoring → model retraining" (each retraining epoch uses 10K samples). However, the paper does not discuss Computational efficiency. How much compute (GPU hours, memory) is required for the full RLVR pipeline?

**Questions:**

1. You report improved performance on non-geometric tasks (MMMU’s Art & Design, Tech & Engineering)—could you clarify: (1) Which subcategories of these tasks benefit most? (2) Do these tasks share structural similarities with geometric reasoning (e.g., visual feature extraction, relational deduction)?
2. Could you conducted any probing experiments (e.g., attention visualization, feature layer analysis) to confirm that the same model capabilities are transferred as a result of training?

---

> ### Author Response · Authors · 2025-11-27
> **R1: Response to Reviewer 183W**
>
> Thank you so much for your valuable suggestions and insightful comments. We deeply appreciate your recognition of the significance and innovative integration of RL/caption/reasoning in our work. We would like to address your concerns as follows:
>
> ### Weakness 1: Dataset Statistics and Categorization
> > The paper presents GeoReasoning-10K as a "high-quality aligned dataset," yet it offers insufficient detail about how well the dataset covers complex geometric edge cases—coverage that is essential for evaluating true model generalization... To address this, the authors should publish a detailed breakdown of the dataset, categorizing samples by geometric type (e.g., 2D vs. 3D, Euclidean vs. non-Euclidean), complexity (e.g., number of components, number of inference steps required), and other relevant dimensions.
>
> Thanks for the insightful suggestion! To address the mentioned concern, we would like to share the statistics of the distribution across different types and difficulties:
>
> | Difficulty | Triangle | Quadrilateral | Pentagon | Circle | Line | Angle | Other | Total |
> | :--- | :--- | :--- | :--- | :--- | :--- | :--- | :--- | :--- |
> | **Easy (1 clause)** | 12.2% | 17.4% | 2.0% | 0.0% | 2.1% | 0.0% | 0.0% | 33.8% |
> | **Medium (2 clauses)** | 5.1% | 6.7% | 0.8% | 1.6% | 1.6% | 0.4% | 2.0% | 18.2% |
> | **Hard ($\ge$3 clauses)** | 10.7% | 15.7% | 1.7% | 6.3% | 4.2% | 1.7% | 7.8% | 48.0% |
>
> As shown in the table above, the easy/medium/hard samples have a proportion of approximately 3:2:5, meaning GeoReasoning-10K places more emphasis on difficult samples, which have at least three clauses to form complex figures.
>
>
> ### Weakness 2: Choice of $\lambda_r$
> > The core innovation of the RLVR phase is the composite reward function $R = \lambda_r \cdot R_{\text{reasoning}} + (1 - \lambda_r) \cdot R_{\text{caption}}$ (with $\lambda_r =0.7$), but the paper fails to validate whether both components are truly necessary—for example, by testing variants that use only one of the two rewards. Moreover, the choice of $\lambda_r = 0.7$ appears arbitrary, as the paper provides no justification or analysis for this hyperparameter setting.
>
> Thanks for the constructive comment. To address the mentioned concern, we have conducted an additional ablation of $\lambda_r$.
>
> | $\lambda_r$ | MathVerse | MathVista |
> | - | - | - |
> | 0 | 27.5% | 48.9% |
> | 0.3 | 26.6% | 49.5% |
> | 0.5 | 26.7% | 49.3% |
> | 0.7 | 27.4% | 50.0% |
> | 1 | 27.5% | 49.8% |
>
> As shown in the table above, $\lambda_r = 0.7$ achieves competitive performance. In addition, introducing the reasoning reward has a major effect on the out-of-domain task (MathVista) and has little effect on the in-domain task of MathVerse.
>
> ### Weakness 3: Computational Cost of RLVR
> > The RLVR Phase involves 5 rounds of "caption generation → reward scoring → model retraining" (each retraining epoch uses 10K samples). However, the paper does not discuss Computational efficiency. How much compute (GPU hours, memory) is required for the full RLVR pipeline?
>
> Thanks for the constructive comment. The overall cost is approximately 13 hours with $8\times$ L20 GPUs. Specifically:
>
> * Question–answer pair generation using the Gemini API takes about 10 hours per 10K samples.
> * Each cold-start epoch takes about 10 minutes.
> * Each RLVR epoch takes about 2–3 hours.
>
> ### Question 1: Explanation of Effectiveness (Task-Level)
> > You report improved performance on non-geometric tasks (MMMU’s Art & Design, Tech & Engineering)—could you clarify: (1) Which subcategories of these tasks benefit most? (2) Do these tasks share structural similarities with geometric reasoning (e.g., visual feature extraction, relational deduction)?
>
> Thanks for the insightful questions!
>
> (1) The most benefited subcategories lie in the questions related to geometric figures. For example:
>
> * For Art & Design, the subcategory is **art theory, design**.
> * For Science, the subcateogry is **Chemistry and Physics**.
> * For Tech & Engineering, the subcategory is **mechanical engineering**.
>
> (2) Yes, all those subcategories of tasks share structural similarities with geometric reasoning, especially in visual feature extraction and mathematical calculation.
>
> ### Question 2: Explanation of Effectiveness (Probing)
> > Could you conducted any probing experiments (e.g., attention visualization, feature layer analysis) to confirm that the same model capabilities are transferred as a result of training?
>
> Thanks for the insightful suggestion. We will provide the attention visualization in the follow-up experiments in the revision.
>
> If you believe our responses have satisfactorily addressed your concerns, we would be most appreciative if you could consider raising the score.
>
> We fully understand the demands on your time and sincerely thank you for your efforts in helping us improve our work. We look forward to any additional feedback you may have.

---

> > ### Comment · Reviewer_183W · 2025-11-28
> >
> > The author has addressed most of my concerns, and I am inclined to maintain my positive score.

---

### Official Review · Reviewer_esbc · 2025-10-31

**Soundness:** 3
**Presentation:** 2
**Contribution:** 3
**Rating:** 4
**Confidence:** 4

**Summary:**

To address the lack of high-quality geometry image-text pair datasets and further enhance the geometric reasoning capabilities of MLLMs, this paper proposes a novel geometric image description generation framework, Geo-Image-Textualization, and releases a high-quality dataset, GeoReasoning-10K. Utilizing the RLVR framework combined with the RAFT method, the authors iteratively optimize both data quality and model capability. Experiments demonstrate that the optimized model exhibits significant improvements across multiple geometric tasks.

**Strengths:**

1. The authors propose a novel iterative "data-model" optimization paradigm, using RL methods to iteratively enhance both dataset and model quality.
2. The experiments are sufficiently comprehensive. The results show that the proposed method not only improves performance on geometric problems but also demonstrates promising enhancements in other out-of-domain areas.

**Weaknesses:**

Although the authors' results show great potential, there are still some limitations:
1. GeoReasoning-10K contains only 10,000 samples, which is significantly smaller than existing synthetic datasets like AutoGeo-100K or GeoPeP-200K. Furthermore, the "image-text alignment" of proposed dataset is not clearly demonstrated (beyond the experimental results). It remains unclear whether the performance improvement stems primarily from data quality or the RLVR process itself; this requires further experimental validation. The authors should apply RLVR optimization to the same AutoGeo-100K or GeoPeP-200K datasets to see if the results can surpass those achieved with the proposed GeoReasoning dataset.

2. The Reasoning Reward relies on an external model to judge answer correctness. The model might generate plausible but actually incorrect descriptions. If the reward model itself has biases or errors in geometric reasoning, these could be amplified; the overall robustness of RLVR needs further substantiation.

If authors can detail these questions, I will consider raise my score.

**Questions:**

1. See weakness.
2. Although optimized via RL, the dataset size is 10K, which remains relatively small compared to other datasets. Further discussion on its efficiency and scalability is needed to convincingly demonstrate the effectiveness of the authors' proposed data generation pipeline.

---

> ### Author Response · Authors · 2025-11-27
> **R1: Response to Reviewer esbc - Part 1**
>
> Thank you so much for your valuable suggestions and insightful comments. We deeply appreciate your recognition of our novel iterative "data-model" paradigm, comprehensiveness of experiments, and effectiveness of the methods in our work. We would like to address your concerns as follows:
>
> ### Weakness 1.1: Scale to Larger Sizes
> > GeoReasoning-10K contains only 10,000 samples, which is significantly smaller than existing synthetic datasets like AutoGeo-100K or GeoPeP-200K.
>
> Thanks for the constructive comment. To address the concern mentioned, we would like to share additional experimental results in Table 4.
>
> | Dataset            | MathVerse | MathVista |
> | ------------------ | --------- | --------- |
> | Geo170K            | 22.0%     | 46.8%     |
> | GeoPep (100K)      | 22.7%     | 47.1%     |
> | GeoGPT4V (23K)     | 24.6%     | 46.0%     |
> | AutoGeo (100K)     | 24.7%     | 46.1%     |
> | GeoReasoning (10K) | **25.8%** | **48.6%** |
>
> As shown in the table, even with this smaller amount of training data, GeoReasoning can still nontrivially outperform other datasets of larger size in MathVerse and MathVista.
>
> The main bottleneck of scaling is the budget, as the process involves a Gemini API call for question–answer pair generation, which incurs substantial cost. However, it is indeed more beneficial to obtain a dataset with more samples, so we scale up the dataset to 20K samples.
>
> The result is shown in the **Figure 13 in our latest manuscript**. The improvement supports both the effectiveness of our pipeline and the superior scaling property of GeoReasoning.
>
> ### Weakness 1.2: Fair Comparison (Baselines without RLVR)
> > Furthermore, the "image-text alignment" of proposed dataset is not clearly demonstrated (beyond the experimental results). It remains unclear whether the performance improvement stems primarily from data quality or the RLVR process itself; this requires further experimental validation.
>
> Thanks for the insightful question. In Table 1 and Table 2, the comparison is already fair since the base model is trained on the provided dataset without RLVR,
> hence the performance improvement solely stems from data quality in those two tables.
>
> ### Weakness 1.3: Fair Comparison (Baselines with RLVR)
> > The authors should apply RLVR optimization to the same AutoGeo-100K or GeoPeP-200K datasets to see if the results can surpass those achieved with the proposed GeoReasoning dataset.
>
> Thanks for the constructive comment. To further address the concern related to fair comparison, we have conducted additional experiments on AutoGeo and GeoPeP with our RLVR process. Two major conclusions can be drawn from the added experiments:
>
> * **1) RLVR significantly improves the performance of datasets on the aligned captioner.** As shown in the table below, the RLVR process provides consistent performance improvements for different datasets on MathVista and MathVerse, demonstrating the effectiveness of this “image–text” alignment paradigm.
>
>
> |  | MathVista | MathVerse |
> | - | - | - |
> | GeoPeP | 47.5% | 24.2% |
> | GeoPeP + RLVR (with aligned captioner)  | 49.1% (+1.6%) | 24.8% (+0.6%) |
> | AutoGeo | 47.8% | 24.6% |
> | AutoGeo + RLVR (with aligned captioner) | 48.7% (+0.9%) | 25.7% (+1.1%) |
> | GeoReasoning | 48.6% | 25.8% |
> | GeoReasoning + RLVR (with aligned captioner) | **50.0% (+1.4%)** | **27.4% (+1.6%)** |
>
> * **2) RLVR also improves the performance of datasets on non-aligned captioners.** It is possible that the improvement originates from the trained captioner model instead of the dataset. Therefore, we further conducted experiments by training the optimized datasets on non-aligned base models. As shown in the table below, the RLVR process still provides consistent improvement after we separate the dataset from its aligned version of the captioner, indicating an improvement in dataset quality.
>
> |  | MathVista | MathVerse |
> | - | - | - |
> | AutoGeo | 47.8% | 24.6% |
> | AutoGeo + RLVR (with base model) | 48.3% (+0.5%) | 25.2% (+0.6%) |
> | GeoPeP | 47.5% | 24.2% |
> | GeoPeP + (with base model) | 48.0% (+0.5%) | 25.0% (+0.8%) |
> | GeoReasoning | **48.6%** | **25.8%** |

---

> > ### Author Response · Authors · 2025-11-27
> > **R1: Response to Reviewer esbc - Part 2**
> >
> > ### Weakness 2: Quality of Reward Signal
> > > The Reasoning Reward relies on an external model to judge answer correctness. The model might generate plausible but actually incorrect descriptions. If the reward model itself has biases or errors in geometric reasoning, these could be amplified; the overall robustness of RLVR needs further substantiation.
> >
> > Thanks for your valuable advice! It is indeed a valid concern. Our reward signal comes from the accuracy gap between good and bad captions. To show that Qwen2.5-7B-Instruct is qualified for this purpose, we compare it with Gemini 2.5-Flash, a well-known model with strong reasoning ability. Specifically, we rerun the rollout stage for 1,000 images, i.e., generating 8 captions for each image with the captioning model (Gemma3-4B). Then we use Qwen2.5-7B-Instruct and Gemini 2.5-Flash to solve the 8,000 question–caption pairs and record the accuracy, as shown below.
> >
> > |  | Qwen2.5-7B-Instruct | Gemini 2.5-Flash |
> > | - | - | - |
> > | Random rollouts | 24.77% | 26.94% |
> >
> > As shown in the table, the gap between Qwen2.5-7B-Instruct and Gemini 2.5-Flash is small for random rollout captions. This suggests that Qwen2.5-7B-Instruct is as good as Gemini 2.5-Flash, at least for the early-stage optimization of the caption.
> >
> > ### Question: Scalability
> > > Although optimized via RL, the dataset size is 10K, which remains relatively small compared to other datasets. Further discussion on its efficiency and scalability is needed to convincingly demonstrate the effectiveness of the authors' proposed data generation pipeline.
> >
> > Please see Weakness 1.1.
> >
> > If you believe our responses have satisfactorily addressed your concerns, we would be most appreciative if you could consider raising the score.
> >
> > We fully understand the demands on your time and sincerely thank you for your efforts in helping us improve our work. We look forward to any additional feedback you may have.

---

### Official Review · Reviewer_owqy · 2025-10-31

**Soundness:** 3
**Presentation:** 2
**Contribution:** 3
**Rating:** 6
**Confidence:** 4

**Summary:**

This paper presents a geometric data generation pipeline with reinforcement learning with verifiable rewards (RLVR). RLVR helps generate captions of higher quality that effectively capture the key features of synthetic geometric images. Models trained on their proposed dataset achieve better performance on in-domain geometric tasks and out-of-distribution math and general tasks.

**Strengths:**

- They propose a geometry dataset with 10,000 image-text pairs.
- They propose an RLVR framework to refine the captions generated for the geometric dataset.
- They not only improve performance on geometric benchmarks, but also enhance performance on standard math tasks and the MMMU task.

**Weaknesses:**

- The writing is a bit confusing, and some parts could be overstated. For example, it says "GeoReasoning-10k achieves non-trivial performance gains" (Ln 022). How can a benchmark have performance gains, and why is the improvement "non-trivial"? They also said the dataset is "fully-aligned" (Ln 082) without statistical support.
- Compared other datasets of 100k and 200k scales (e.g., AutoGeo-100k, GeoPeP), it only has 10k data which is much smaller. Since the data is very effective and generated by an automatic pipeline, it would be better to scale much of the other data.
- It lacks detailed analyses. Naturally, training on 10,000 synthetic domain-specific data can not help out-of-distribution performance that much. It would be very interesting to see the reason behind it.
- It is concerning that for the geometry tasks, training on their GeoReasoning only has marginal improvement.

**Questions:**

- Among all the generated data, what is the result of human validation? How is the hallucination rate?
- What is the computation and time used for data generation? Is it possible to upscale it to 100k? At that scale, would it still help performance on out-of-distribution benchmarks?
- How can training on your geometric synthetic data help performance on other tasks? On what cases does it improve, and on what cases does it drop?
- Compared to other geometric datasets, after training on georeasoning, what are the advantages and disadvantages of the geometry split?

---

> ### Author Response · Authors · 2025-11-27
> **R1: Response to Reviewer owqy - Part 1**
>
> Thank you so much for your valuable suggestions and insightful comments. We deeply appreciate your recognition of the effectiveness and good generalization performance of our work. We would like to address your concerns as follows:
>
> ### Weakness 1: Clarification of Main Contribution
> > The writing is a bit confusing, and some parts could be overstated. For example, it says "GeoReasoning-10k achieves non-trivial performance gains" (Ln 022). How can a benchmark have performance gains, and why is the improvement "non-trivial"? They also said the dataset is "fully-aligned" (Ln 082) without statistical support.
>
> Thanks for the question. **GeoReasoning-10K is a training dataset**, not a benchmark, thus it provides performance gains when the model is fine-tuned on this dataset. We will clarify this more in the revision to avoid confusion.
>
> The performance gains are nontrivial because most of the +2.8%–4.8% improvement over the base model is at least one standard deviation away, meaning these improvements are highly unlikely to be caused by variance.
>
> The term fully aligned refers to the alignment process of RLVR. To support this argument, we conducted an additional human validation on 100 randomly selected samples, where all samples have an aligned image and text description. We will include these results in the revisions as well. Thanks for the suggestion!
>
> ### Weakness 2: Scale to Larger Sizes
> > Compared other datasets of 100k and 200k scales (e.g., AutoGeo-100k, GeoPeP), it only has 10k data which is much smaller. Since the data is very effective and generated by an automatic pipeline, it would be better to scale much of the other data.
>
> Thanks for the suggestion. The main bottleneck of scaling is the cost, as the process involves a Gemini API call for question–answer pair generation, which incurs substantial cost. However, it is indeed more beneficial to obtain a dataset with more samples, so we scale up the dataset to 20K samples.
>
> The result is shown in the **Figure 13 in our latest manuscript**. The improvement supports both the effectiveness of our pipeline and the superior scaling property of GeoReasoning.
>
> In addition, even with this smaller amount of training data, GeoReasoning can still nontrivially outperform other datasets of larger size in MathVerse and MathVista, as shown in the table below.
>
> | Dataset            | MathVerse | MathVista |
> | ------------------ | --------- | --------- |
> | Geo170K            | 22.0%     | 46.8%     |
> | GeoPep (100K)      | 22.7%     | 47.1%     |
> | GeoGPT4V (23K)     | 24.6%     | 46.0%     |
> | AutoGeo (100K)     | 24.7%     | 46.1%     |
> | GeoReasoning (10K) | **25.8%** | **48.6%** |
>
> ### Weakness 3: Explanation of the Effectiveness
> > It lacks detailed analyses. Naturally, training on 10,000 synthetic domain-specific data can not help out-of-distribution performance that much. It would be very interesting to see the reason behind it.
>
> Thanks for the valuable advice. To address this concern, we have conducted two additional analyses at both the task level and the case level to understand GeoReasoning's effectiveness.
>
> At the task level, we find that the improved tasks are mostly related to geometric figures, especially those that **rely on visual feature extraction and mathematical calculation**. For example:
>     - For Art & Design, the most benefited subcategory is art theory, design.
>     - For Science, the most benefited subcategory is Chemistry and Physics.
>     - For Tech & Engineering, the most benefited subcategory is mechanical engineering.
>
> At the case level, we manually sampled improved cases and found that the **visual extraction and analysis capacity** is largely improved by training on GeoReasoning, which can be transferred to out-of-domain tasks. For example:
>
> ```
> # ID: 5
>
> Answer: Let's analyze the given figure. ...We are given the following angles:\n $\angle E = x^\circ$, $\angle F = (x+20)^\circ$, $\angle G = (x+5)^\circ$, $\angle H = (x-5)^\circ$, $\angle J = (x+10)^\circ$ ... Now we can find the measure of $\angle H$:\n $\angle H = (x-5)^\circ = (102 - 5)^\circ = 97^\circ$\n\nTherefore, $m\angle H = 97^\circ$.\n\nFinal Answer: The final answer is A.\n
> ```
>
> This leads to out-of-domain generalization in similar aspects. For instance:
>   - In Figure 7, it helps analyze the shape of the mechanical part to find the correct left view.
>   - In Figure 10, it helps extract the height of each bar in the chart and translate it into the precipitation value.
>
> In short, the success on the out-of-domain cases can be viewed as an extension of the in-domain cases, which originates from the improvements in visual feature extraction and mathematical reasoning.

---

> > ### Author Response · Authors · 2025-11-27
> > **R1: Response to Reviewer owqy - Part 2**
> >
> > ### Weakness 4: In-Domain Performance
> > > It is concerning that for the geometry tasks, training on their GeoReasoning only has marginal improvement.
> >
> > Thanks for the insightful comment. It is mainly because the base model is already strong on the geometry tasks. The <2.0% improvements of AutoGeo and GeoPeP also demonstrate this phenomenon.
> >
> > In addition, GeoReasoning focuses more on out-of-domain generalization ability, which is the main contribution of this dataset.
> >
> > ### Question 1: Human Validation
> > > Among all the generated data, what is the result of human validation? How is the hallucination rate?
> >
> > Thanks for the insightful question. To address this concern, we have conducted an additional human validation, where 100 captions were randomly sampled from GeoReasoning, and **no hallucination was found after manual checking in this sampled subset**.
> >
> > ### Question 2: Data Generation Cost
> > > What is the computation and time used for data generation? Is it possible to upscale it to 100k? At that scale, would it still help performance on out-of-distribution benchmarks?
> >
> > Thanks for the constructive comment. The overall cost is approximately 13 hours with $8\times$ L20 GPUs. Specifically:
> >
> > * Question–answer pair generation using the Gemini API takes about 10 hours per 10K samples.
> > * Each cold-start epoch takes about 10 minutes.
> > * Each RLVR epoch takes about 2–3 hours.
> >
> > The main bottleneck of scaling is the budget, as the process involves a Gemini API call for question–answer pair generation, which incurs substantial cost when scaling up to 100K samples. We will investigate approaches in the future for reducing the cost. Thanks for the suggestion!
> >
> > ### Question 3: Explanation of the Effectiveness II
> > > How can training on your geometric synthetic data help performance on other tasks? On what cases does it improve, and on what cases does it drop?
> >
> > Thanks for the insightful question. To address the concern, we have manually checked cases in both improved and degraded tasks and found two conclusions.
> >
> > * (1) Improved cases mostly lie in the questions related to geometric figures, especially those that **rely on visual feature extraction and mathematical calculation**. For example:
> >
> >     - For Art & Design, the most benefited subcategory is **art theory, design**.
> >     - For Science, the most benefited subcateogry is **Chemistry and Physics**.
> >     - For Tech & Engineering, the most benefited subcategory is **mechanical engineering**.
> >
> > * (2) Despite the improvement, we find that our model has minor degradation on cases that **largely depend on knowledge**, such as
> >     - Humanities & Social Science, where almost all geometry datasets have performance drop according to Table 2.
> >
> > ### Question 4: Explanation of the Effectiveness (Baseline comparison)
> > > Compared to other geometric datasets, after training on georeasoning, what are the advantages and disadvantages of the geometry split?
> >
> > Thanks for the insightful question. By training on GeoReasoning, our model shows **superior visual information extraction capability** compared to other datasets, at the cost of **minor degradation in knowledge**. This is similar to the response to Question 3.
> >
> > In addition, by manually checking the advantageous and disadvantageous cases, we find that the model trained on GeoReasoning can quickly determine geometric relations and output straightforward calculation formulas, while the model trained on AutoGeo cannot precisely establish geometric relations, causing uncertainty in the computations.
> >
> > If you believe our responses have satisfactorily addressed your concerns, we would be most appreciative if you could consider raising the score.
> >
> > We fully understand the demands on your time and sincerely thank you for your efforts in helping us improve our work. We look forward to any additional feedback you may have.

---

### Author Response · Authors · 2025-12-03
**Rebuttal Summary**

Dear SAC, AC, and Reviewers,

We sincerely appreciate the time and effort you have invested in reviewing our manuscript and providing such insightful suggestions and comments. We are pleased to note that the reviewers acknowledged that our paper:

* **Proposes RLVR to improve image-caption alignment in geometric problems** (owqy, esbc, 183W, hdrJ)
* **Provides generalizable performance boost in out-of-domain tasks** (owqy, esbc, 183W, hdrJ)
* **Provides comprehensive experiments to support the claim** (esbc, 183W, hdrJ)

To save time for every reviewer, AC, and SAC, we would like to share a brief summary of the rebuttal and the relevant concerns raised during the discussion.

* Updated ratings
  - Reviewer owqy: 6
  - Reviewer 183W: 6
  - Reviewer esbc: 4
  - Reviewer hdrJ: 2 $\rightarrow \ge 4$ (see email)

* Relevant concerns **addressed during the discussion**

| Concern                    | Raised by reviewer |
| -------------------------- | ------------------ |
| Scale up to a larger-size dataset | owqy, esbc |
| Explanation of the effectiveness  | owqy, 183W |
| Copmutational Cost                | owqy, 183W |
| Fair comparison                   | esbc |
| Verification of reward signal quality | esbc |
| Human validation                  | owqy |
| Data generation cost              | owqy |
| Data statistics                   | 183W |
| Ablation studies                  | 183W |
| More baselines                    | hdrJ |
| Clarification on Figure 1         | hdrJ |
| Clarification on Data Generation Pipeline | hdrJ |

In addition, we have incorporated the larger-size experiments in Table 8 and Figure 13 (Appendix F) in the updated revision.

Thank you very much for reviewing the comments, and we hope this summary can help reduce the workload.

Sincerely,

Paper 13527 Authors

---

### Meta-Review · Area_Chair_yBjx · 2026-01-08

**Summary:**

The paper’s main strength is that it introduces a new dataset and paradigm that can iteratively improve both data quality and model capability, and is intended to support reinforcement-learning-style methods. Reviewers acknowledge that the work contributes to the field and that the proposed paradigm has some novelty, but they also highlight clear limitations.

In particular, three reviewers (including one who gave a positive score) argue that the claimed advantages of the dataset and paradigm are either obvious or overstated, and not fully convincing—suggesting that similar gains might be achievable with existing datasets or alternative data collection strategies. In other words, the contribution does not appear to be truly breakthrough. Additionally, one reviewer notes that the dataset may be too small, with an insufficient number of samples to strongly support the paper’s claims.

**Reviewer Concerns:**

In my view, the rebuttal makes the claimed contributions and strengths of the work more credible and less overstated.
 However, it does not fully resolve the concern that the dataset is too small, nor does it convincingly address the broader criticism that the contribution is not sufficiently breakthrough.

**Reviewer Scores:**

In my view, if the reviewers had been able to participate fully in the discussion, Reviewer hdrJ might have increased their score from 2 to 4.

---

### Decision · Program_Chairs · 2026-01-26

Reject